# Development of Palm Fatty Acid Distillate-Containing Medium for Biosurfactant Production by *Pseudomonas* sp. LM19

**DOI:** 10.3390/molecules24142613

**Published:** 2019-07-18

**Authors:** Abdul Hamid Nurfarahin, Mohd Shamzi Mohamed, Lai Yee Phang

**Affiliations:** 1Department of Bioprocess Technology, Faculty of Biotechnology and Biomolecular Sciences, Universiti Putra Malaysia, 43400 UPM Serdang, Selangor, Malaysia; 2Bioprocessing and Biomanufacturing Research Centre, Faculty of Biotechnology and Biomolecular Sciences, Universiti Putra Malaysia, 43400 UPM Serdang, Selangor, Malaysia

**Keywords:** biosurfactants, palm fatty acid distillate, *Pseudomonas* sp., medium development, statistical technique

## Abstract

High production costs of biosurfactants are mainly caused by the usage of the expensive substrate and long fermentation period which undermines their potential in bioremediation processes, food, and cosmetic industries even though they, owing to the biodegradability, lower toxicity, and raise specificity traits. One way to circumvent this is to improvise the formulation of biosurfactant-production medium by using cheaper substrate. A culture medium utilizing palm fatty acid distillate (PFAD), a palm oil refinery by-product, was first developed through one-factor-at-a-time (OFAT) technique and further refined by means of the statistical design method of factorial and response surface modeling to enhance the biosurfactant production from *Pseudomonas* sp. LM19. The results shows that, the optimized culture medium containing: 1.148% (*v*/*v*) PFAD; 4.054 g/L KH_2_PO_4_; 1.30 g/L yeast extract; 0.023 g/L sodium-EDTA; 1.057 g/L MgSO_4_·7H_2_O; 0.75 g/L K_2_HPO_4_; 0.20 g/L CaCl_2_·2H_2_O; 0.080 g/L FeCl_3_·6H_2_O gave the maximum biosurfactant productivity. This study demonstrated that the cell concentration and biosurfactant productivity could reach up to 8.5 × 10^9^ CFU/mL and 0.346 g/L/day, respectively after seven days of growth, which were comparable to the values predicted by an RSM regression model, i.e., 8.4 × 10^9^ CFU/mL and 0.347 g/L/day, respectively. Eleven rhamnolipid congeners were detected, in which dirhamnolipid accounted for 58% and monorhamnolipid was 42%. All in all, manipulation of palm oil by-products proved to be a feasible substrate for increasing the biosurfactant production about 3.55-fold as shown in this study.

## 1. Introduction

The demand for surfactants in various industries has increased each year to the tune of almost USD 24 million in revenues in 2009 and is predicted to reach USD 2.8 billion by 2023 [1]. This can be attributed to numerous advantages provided by biosurfactants over synthetic surfactants in the surface-active compound market such as biodegradability, lower toxicity and well-functionality in extreme environments. Biosurfactants are organic compounds that consist of both hydrophilic and hydrophobic moiety produced by a different class of microorganisms including bacteria, yeast and fungi with a purpose of reducing the surface tension or interfacial tension between two different liquids [2]. They are widely used in agriculture for improving plant growth [3], aiding gene therapy procedure in pharmaceuticals industry [4] and also providing a low irritancy effect in skincare products in the cosmetic industry [5].

Though biosurfactants are deemed advantageous over chemical surfactants, the use of the expensive substrate, low production yield, poor extraction and purification methods eventually lead to higher overall production cost, thus confounding their potential for transitioning to larger commercial-scale production. Current research in solving these problems among others are focusing on the utilization of substrate from the by-products of other bioprocesses, optimization of environmental conditions involved in the production process and minimizing the existing steps in extraction and purification process. In spite of these efforts, biosurfactant yield remains relatively low from an economic point of view [6]. To drive the cost further down, more systematic approaches of manipulating cheaper raw material as a substrate for biosurfactant-production medium with the aid of computational optimization tools have risen in recent years due to the wider availability of powerful simulation packages to design experiments with predictive mathematical techniques such as Plackett–Burman design (PBD) and response surface methodology (RSM) [7,8].

Wastes containing a high amount of free fatty acids (FFA) in food and agricultural sectors have a high potential to be substrates in biosurfactant production since they are low-cost, normally found in abundance and readily available most of the time. For example, waste frying oil [9], waste frying coconut oil and olive oil mill waste [10] all contain a high amount of FFA proved to be a suitable carbon source in rhamnolipid production by *Pseudomonas* sp. Malaysia has reclaimed its position as the world’s major exporter of palm oil to China since 2016 and this has caused a tremendous amount of palm oil by-products generated from this industry. Palm fatty acid distillate (PFAD) that is produced from the palm oil refining process in the mill has a yellowish appearance and forms a solid when left at room temperature. Information and research with regards to PFAD are still limited due to its presumed worthlessness. PFAD consists almost 70 to 90% of FFA. This property could very well address the aforementioned two key issues in biosurfactant production by lowering the production cost in which PFAD could be optional substrate and possibly increasing the biosurfactant yield due to the existence of FFA in culture medium, thus adding the value to the palm oil industry wastes in Malaysia. To date, very few studies have reported the utilization of PFAD for biosurfactant production.

Apart from the incorporation of PFAD into the production medium, other complementary nutritional components should be tweaked as well to tailor the preference of different types of biosurfactant-producing microorganisms. These nutritional components are usually categorized into macronutrients (carbon and nitrogen source) and micronutrients (minerals, vitamins and trace elements). Optimization of these factors through proper statistical techniques is one of the sound approaches to tackle the issue of overpricing in biosurfactant production. Additionally, the identification of the biosurfactants class with its congeners is also very significant since studies made were few for certain biosurfactant type [11]. The novelty of this work comes from the utilization of PFAD as a substrate in medium developed through both combinations of traditional and statistical tools for biosurfactant production by *Pseudomonas* sp., which has never done before to our knowledge. In addition, the importance of identifying biosurfactant class and congeners was discussed in this work.

Therefore, the main goal of this study was to develop a production medium comprised of necessary nutritional components to produce biosurfactant by *Pseudomonas* sp. LM19 with PFAD as the main substrate using combined methods of one-factor-at-time (OFAT) and statistical approaches: PBD and RSM. The second goal is to identify the biosurfactant class produced with its congeners in optimized production medium. 

## 2. Results and Discussion

### 2.1. Free Fatty Acid (FFA) Compositions in PFAD 

PFAD was originally a by-product from the refining process of crude palm oil. The GC-MS analysis shows the compositional profile of FFA ranging from C_12_ to C_18_ (Table 1) in which the major fatty acid constituents consist of palmitic acid (50.30%) and oleic acid (28.09%). 

A report by Jumaah et al. [12] also supported this finding by stating that the major FFA components of Malaysian derived PFAD are palmitic acid (48.90%) and oleic acid (37.40%). On the contrary, different major FFA components in PFAD were reported by Nazren Radzuan et al. [13] comprising of oleic acid (41.86%), pentadecanoic acid (17.98%), tridecylic acid (13.98%) and palmitic acid (13.62%). This discrepancy was most probably due to PFAD being a complex by-product. Its components might be varied according to various factors such as the refining conditions in between batches, the source of crude palm oil obtained and the inherent seasonal characteristics of the crude palm oil itself. Since much earlier research proved that FFA-containing substrates were among the finest carbon sources for biosurfactant production by *Pseudomonas* sp., [14], PFAD had been selected to be the sole carbon source in this study.

### 2.2. Selection of Production Medium

Among the five different production media (Luria Bertani (LB), Bushnell Haas (BH), modified BH, Mineral Salt (MS) and Phosphate-limited Proteose-peptone Ammonium Salt (PPAS)) (Table 2), the highest cell growth was observed in LB medium (1.7 × 10^10^ CFU/mL), whereas the lowest cell biomass was generated in PPAS medium (2.2 × 10^8^ CFU/mL). On the other hand, *Pseudomonas* sp. LM19 produced the maximum biosurfactant in modified BH medium (0.68 g/L) with maximum surface tension (33.11 mN/m) which records the highest E24 (59.62%) compared to the original BH medium and the rest. MS medium otherwise indicated a comparable fermentation performance in terms of cell growth but not quite competitive enough to produce biosurfactant as the modified BH medium.

From these results, it shows that the LB medium favours the cell formation pathways of *Pseudomonas* sp. LM19 as opposed to the modified BH medium which initiates the pathway towards producing more biosurfactants. It is opined here that some of the components in LB medium, possibly that of yeast extract, whereby having proteins, carbohydrates, nucleic acid, free amino acids, short chain of peptides, and trace elements in its composition was an obvious choice of nitrogen source to aid in the build-up of some components for the cells themselves compared to the formation of other secondary metabolites [15]. 

Original BH medium having low production of biosurfactant (0.05 g/L) underwent modification by the addition of sodium EDTA (C_10_H_16_N_2_O_8_). This salt primarily functions as a chelating agent to solubilize Fe^3+^ ion in the medium so that mineral consumption by *Pseudomonas* sp. LM19 could be further enhanced when present in a small amount. The presence of Fe^3+^ ion alone sans the chelating agent in the medium causes them to precipitate and became insoluble even after sterilization took place [16]. The addition of chelating agent boosted the biosurfactant production up to 13.6 fold. Poor overall performance by PPAS medium might be due to the excessive chloride ion (Cl^−^) presented in the medium compared to the others. Chloride ion is associated with the salinity level of the medium. High salinity level of the medium could lead to the imbalance of osmotic pressure in between extracellular and intracellular environments, which leads to the cell lysis. As for a modified BH medium, it has a sufficient concentration of buffer system (KH_2_PO_4_ and K_2_HPO_4_) to elevate the cultural pH. Nitrogen source (NH_4_NO_3_) aids in the formation of protein and nucleic acids in the cells, minerals such as CaCl_2_·2H_2_O acts as a common mediator in signal delivering processes from the cell surface [17] and MgSO_4_·7H_2_O for energy generation [18], while metal ion (FeCl_3_·6H_2_O) as a cofactor for the production of physiologically active materials i.e., biosurfactants [19]. All of these components satisfied the strategies in designing the media as suggested by Singh et al. [20]. On the other hand, the deficiency of MS medium might due to the minimal amount of specific micro and macronutrients found in the medium leading to the only average amount of biosurfactant produced. Based on the final amount of biosurfactant produced, modified BH medium was selected from the five possible basal media as the subject for further study in optimization.

### 2.3. Screening Procedure: One-Factor-at-A-Time (OFAT)

#### 2.3.1. Inoculum Size

It was observed that biosurfactant production slightly increased when the percentages of inoculum size used were doubled and even quadrupled from the initial 1% (*v*/*v*). Figure 1a demonstrated a noticeable increase of about 73.98% in biosurfactant production (1.19 g/L) with 67.95% of E24 when 6% (*v*/*v*) of inoculum size was used. In terms of cell generation, it was immediately enhanced to 62.07% (9.4 × 10^9^ CFU/mL) upon doubling the inoculum size to 2% (*v*/*v*) from the initial condition. However, a further increase in inoculum size seemingly caused a further reduction in cell concentration, be it in terms of colony forming unit or optical density.

The biosurfactants readings were produced between Day 8 to Day 9 of the incubation period (late stationary phase) pointing to a secondary metabolite secretion and the activation of the quorum sensing system in relation to accumulated cell population in culture. Theoretically, high cell population is directly linked to the activation phase of rhamnolipid by the microorganisms [21,22] since high cell density is required to produce a high amount of biosurfactant. A slight decrease in biosurfactant production at higher inoculum size might be attributed to the ongoing competition between individual bacterial cells of bacteria for nutrients in the medium. Patil et al. [23] and Lan et al. [24] both found that the inoculum size of 4% and 2% (*v*/*v*) of *Pseudomonas* sp., respectively, to be the most favourable for producing rhamnolipid and reportedly experiencing inhibiting effects from any inoculum sizes set higher or lower than the prescribed range for their production media. Thus, the 6% (*v*/*v*) inoculum size promoting in excess of 1.00 g/L of biosurfactant production was selected for further optimization of other process parameters. 

#### 2.3.2. PFAD Concentrations

According to Figure 1b the amount of biosurfactant peaked when PFAD concentration was increased from 0.5% to 1% (1.19 g/L) with 8.03 × 10^9^ CFU/mL of the cell. The amount of biosurfactant produced then drastically dropped when a higher amount of PFAD was added into the production medium. The lowest biosurfactant production could be observed in the production media containing 0.5% and 6% (*v*/*v*) of PFAD. Interestingly, high cell density (≈10^9^ CFU/mL) was still being generated when using a higher concentration of PFAD.

*P. aeruginosa* possesses the capability to metabolize different FFA chain length (C_4_-C_18_) [25] which presented in the PFAD substrate. At higher concentrations of PFAD used in this study (more than 1% (*v*/*v*)), microorganisms produce lipase to digest PFAD into a various chain of FFA and cause it to further accumulate without being fully consumed by the microbe itself, leading to the decrement of pH in the culture broth. On the other hand, high cell density still being produced in the production medium containing a higher concentration of PFAD (2%, 4% and 6%) (*v*/*v*) which lead to the facts that metabolites other than biosurfactants might be generated in this process. In contrast, a lower concentration of PFAD, 0.5% (*v*/*v*) leads to an insufficient amount of carbon source required for cell formation and biosurfactant production during the fermentation process. 

An earlier study by Nazren Radzuan et al. [13] manipulated 20, 50 and 100 g/L of PFAD to produce biosurfactants from *P. aeruginosa* PAO1. The maximum amount of rhamnolipid (0.43 g/L) was achieved with no significant difference in the growth and rhamnolipid production with other PFAD concentrations tested which actually contradicted from this study. This happened due to the availability of PFAD in culture media, which is not directly related to the total initial concentration of the PFAD, due to the low solubility of PFAD and heterogeneous system, which suggests that roughly equal amounts of PFAD were dissolved even for different initial concentrations, whereas, in this study, liquefied PFAD (after heating) was used by adding drop by drop into the culture medium with means to increase the surface availability of the substrate towards microbes to produce biosurfactant and to improve the solubility and homogeneity of the system. On the other hand, this trend can be seen when *Haloarcula* sp. IRU1 producing the lowest amount of biosurfactant at 0.5% (*v*/*v*) of olive oil and produced the highest amount of biosurfactant when supplemented with olive oil at 4% (*v*/*v*) [26]. Therefore, a 1% (*v*/*v*) of PFAD was used for further optimization of different parameters.

#### 2.3.3. Type of Nitrogen Sources

Six different nitrogen sources (three organic and inorganic nitrogen, each) with similar nitrogen content (0.35 g/L) were tested in this work to determine the most suitable one to further enhance the biosurfactant production and cell formation by *Pseudomonas* sp. LM19. In this study, both types of nitrogen sources were capable of causing the production of biosurfactant and high cell density generation with the exception of ammonium sulphate. Among organic nitrogen sources, yeast extract demonstrated the highest increment of biosurfactant production (26.05%) together with the cell formation (8.80 × 10^9^ CFU/mL) on the 7th day of incubation as shown in Figure 1c. The lowest biosurfactant production (0.01 g/L) was observed in the medium containing ammonium sulphate.

In the previous study, yeast extract proved to boost the production of biosurfactant and cell [15,27]. Yeast extract has an amalgamation of proteins, carbohydrates, short peptide chains, free amino acids and nucleic acids which make it a rich source of nitrogen and essential trace elements vital for cell growth and product formation. On the other hand, lower production of biosurfactant caused by the addition of ammonium sulphate was similar to the work by Rashedi et al. [28]. Additionally, nitrate-based nitrogen sources may be attributed to the more readily available and simple nitrogen components to be consumed in culture compared to the non-nitrate based nitrogen sources [29]. On the contrary, *P. putida* MTCC 2467 was capable of utilizing ammonium sulphate well in yielding a higher biosurfactant amount when compared to other nitrogen sources [30]. In short, the preference of nitrogen sources used, which leads to a higher amount of biosurfactant produced, depends solely on the microbes used. Therefore, yeast extract was selected as the nitrogen source for the following OFAT screening. In this work, oxygen was supplied by the diffusion through the cotton plug of shake flask as the respiratory activity of microorganism and environmental conditions do not, in general, change rapidly and can be regarded as in steady state [31]. At the maximum possible operating conditions (shaking frequencies below 350 rpm) and high volumes of fill (100 cm^3^), the metabolic activity of the microorganisms is limited by the transport processes in the shaking flask. Therefore, the working volume used was the maximum 40% from the total volume of shake flask ensuring high surface area for maximum gas/liquid mass transport as suggested by Ryan [32].

#### 2.3.4. Yeast Extract Concentrations

According to Figure 1d, the usage of 1.5 g/L of yeast extract gave the highest reading (1.79 g/L) of biosurfactant. However, all concentrations of yeast extract tested generated a high amount of cells with the highest (1.06 × 10^10^ CFU/mL) can be seen in medium containing 4.5 g/L of yeast extract. The least amount of biosurfactant (0.05 g/L) could be observed when 6.0 g/L of yeast extract was utilized. Different concentrations of yeast extract, 0.5, 1.5, 3.0, 4.5 and 6.0 g/L corresponding to the C/N ratio of 20.00, 6.67, 3.33, 2.22 and 1.67, respectively. Basically, the C/N ratio is very important as it will affect the levels of the transcripts of certain genes, which is required for the uptake and metabolism of carbon sources and the formation of by-products [33]. Meanwhile, Nurfarahin et al. [29] reported that *Pseudomonas* sp. yielded a maximum amount of biosurfactant at C/N ratio 6 to 13 based on most of the previous works. This was further being supported by this work in which the C/N ratio of 6.67 demonstrated the maximum amount of biosurfactant produced at the nitrogen-limiting condition. Wu et al. [34] also concurred that the maximum biosurfactant occurred at a C/N ratio 52 (nitrogen limiting condition) when glycerol was used as carbon source, but markedly decreased as the C/N ratio was increased or decreased from 52. It showed that a lower C/N ratio in which the biosurfactant production increased under the limitation of nitrogen rather than carbon sources. These results proved that higher C/N ratios (i.e., low nitrogen levels) inhibit the bacterial growth, favouring cell metabolism towards the production of metabolites like biosurfactants [29]. Therefore, 1.5 g/L of yeast extract was carried out into the following PBD screening of significant medium components. 

### 2.4. Screening of Significant Medium Components: Plackett–Burman Design (PBD)

Statistical approaches are powerful tools for identifying the main factors from a multivariable system and reducing the error to determine the effect of each category. PBD is one of the statistical approach design used to screen significant factors among other potential factors involved. In this approach, mostly only main effects were measured. In PBD, data generated were used to identify the few significant factors which can give the highest value of the response. Then, different levels of the selected factors were assessed to obtain the optimum level while the interactions among the factors in the system were completely ignored. 

In this work, PBD was employed to analyse the effect of seven different medium components (factors) which included macronutrient; A: KH_2_PO_4_, B: K_2_HPO_4_, C: MgSO_4_·7H_2_O, D: CaCl·2H_2_O, E: yeast extract and micronutrient F: FeCl_3_·6H_2_O, G: sodium-EDTA in which were modified according to OFAT’s finding on the cell growth and maximum biosurfactant productivity by *Pseudomonas* sp. LM19. The screening of significant medium components to produce higher biosurfactant productivity and cell growth of *Pseudomonas* sp. LM 19 was conducted by preparing several media with varying concentrations of the components as suggested by the PBD (Table 3). For 10 days of incubation, maximum cell concentration and biosurfactant productivity taken on a particular incubation time was used as a response. It was observed that there were differences in biosurfactant productivity in the range of 0.0051 to 0.2722 g/L/day and cell concentration in the range of 7.3 × 10^9^ to 14.2 × 10^9^ CFU/mL (Table 3). Analysis of variance (ANOVA) was performed to find the effect and the contribution for each factor.

According to ANOVA test (Appendix A), the results showed that the presence of high level of KH_2_PO_4_ and yeast extract in the production medium positively affected the cell growth, whereas sodium-EDTA gave negative effects to cell formation when present at a high level. As for biosurfactant, high levels of KH_2_PO_4_ and MgSO_4_·7H_2_O gave positive effects on maximum biosurfactant productivity while the inverse occurred when yeast extract was present at a high level of concentration. The correlation coefficients (R^2^) for cell formation and maximum biosurfactant productivity were found to be 0.8602 and 0.8615, respectively, showing good fitness of the model. It was given that “Predictive R-squared” were 0.8135 and 0.8615 for cell formation and maximum biosurfactant productivity, respectively, which concluded that these values were in reasonable agreement with the “Adjusted R-squared” of 0.7004 and 0.7111 (difference less than 0.2), demonstrated that a good agreement between the experimental and predicted values for both cell growth and maximum biosurfactant productivity. Therefore, four factors (KH_2_PO_4_, yeast extract, MgSO_4_·7H_2_O and sodium-EDTA) had been subjected to RSM treatment for modification of production medium towards yielding higher biosurfactant and cell concentration by *Pseudomonas* sp. LM19.

### 2.5. Optimization Procedure: Response Surface Methodology (RSM)

#### 2.5.1. Experimental Data for RSM

For further optimization of significant medium components using a statistical approach, RSM based on central composite design (CCD) was employed for the next stage. A central composite design was created based on five factors (four factors from PBD and one factor was the sole carbon source, PFAD) with the cell growth, *Y*_1_ and the maximum biosurfactant productivity, *Y*_2_ as responses. The responses from 53 experimental runs with three sets of blocking are presented in Table 4. Maximum cell growth (9 × 10^9^ CFU/mL) was achieved in Run 27 medium comprising of 1% (*v*/*v*) PFAD, 4.10 g/L KH_2_PO_4_, 1.30 g/L yeast extract, 0.03 g/L sodium-EDTA and 1.00 g/L MgSO_4_·7H_2_O which corresponded to biosurfactant productivity of 0.3473 g/L/day (Day 6). In addition, the highest biosurfactant productivity (0.3705 g/L/day) can be observed in Run 31 (Day 6) having comparable cell concentration of 8.9 × 10^9^ CFU/mL to Run 27. The cell concentration and biosurfactant productivity peaked at similar medium conditions (Run 27 and Run 31) suggested that the requirement of similar nutritional conditions was expressed for both cellular growth and metabolite production, especially the secondary metabolites like biosurfactants in the stationary phase.

#### 2.5.2. Regression Model for Cell Growth and Maximum Biosurfactant Productivity

Data in Table 4 were analysed for multiple regression analysis using least squares regression to fit into quadratic (second order) polynomial model for cell growth, *Y*_1_ and maximum biosurfactant productivity, *Y*_2_. The initial result of the ANOVA test (Appendix A) indicated that nine interactive terms were to be excluded from the model describing cell growth since their contribution was not at least 95% significant. In addition, it shows that only yeast extract (C), sodium-EDTA (D) and MgSO_4_·7H_2_O (E) were significant terms contributing to the model with the *ρ*-value less than 0.05. The interactions between the different medium components only can be found between yeast extract and sodium-EDTA (C*D) with more than 95% significance.

The model equation is presented in uncoded units as Equation (1). Besides that, R^2^ (coefficient of determination) value obtained from regression equation (Equation (1)) was 0.8679, showing that 86.8% of the variability in the cell growth, *Y*_1_ response can be explained by the second order polynomial model equation (Equation (1)). The ‘R^2^ predictive’ of 0.7369 also was in reasonable agreement with the ‘R^2^ adjusted’ of 0.8389 with the difference less than 0.2. However, the value of ‘R^2^ adjusted’ might not explain well the data variation in the region of experimentation to demonstrate that the model is well-fitted. Therefore, the lack of fit test was used to support the competency of the fitted model.
(1)$$\begin{array}{c} Cell\;growth,\;Y_{1}=8.55-0.22\;yeast\;extract-0.23\;sodium•EDTA-0.34\;MgSO_{4}\cdot \\ 7H_{2}O-0.23\;yeast\;extract*sodium•EDTA-0.53PFAD*PFAD-0.31KH_{2}PO_{4}*\\ KH_{2}PO_{4}-0.87\;yeast\;extract*yeast\;extract-0.35\;sodium•EDTA*sodium•\\ EDTA-0.73\;MgSO_{4}\cdot 7H_{2}O*MgSO_{4}\cdot 7H_{2}O. \end{array}$$

In this work, yeast extract was employed as the main nitrogen source for the growth of *Pseudomonas* sp. LM19. This linear term was being mentioned in Equation (1). The model may also face the reductive effect of its quadratic term indicating potential lethal effects possibly happening at higher concentrations. When yeast extract was used at a high concentration with constant PFAD (carbon source) concentration, it leads to the lower C/N ratio that might not favourable for the growth of the cell. This pattern also had been reported by Chang et al. [35], in which higher concentrations of nitrogen source in the culture medium cause the inhibition of *Cryptococcus* sp. growth. Conversely, a low nitrogen source (0.1%) in medium might not significantly affect cell mass, but it will inhibit lipid accumulation. The sodium-EDTA and MgSO4·7H_2_O term also were significantly affecting cell growth during the fermentation process and showing a negative effect when both compounds were present in high concentrations. Sodium-EDTA was reported to cause the disruption of the cell wall of Gram-negative bacteria when present in excess [36], while MgSO_4_·7H_2_O is the metal ion that is required to synthesize the energy for cell growth and might be an intracellular threat when present in excess [37]. Figure 2a shows that the response surface generated using Equation (1) points to an interactive effect between yeast extract and sodium-EDTA towards cell growth during biosurfactant production. Each figure demonstrated the effect of changing those two variables (yeast extract and sodium-EDTA) on cell formation during fermentation by *Pseudomonas* sp. LM19. Optimization simulation of the model predicted that maximum cell concentration can be achieved at 8.396 × 10^9^ CFU/mL.

As for maximum biosurfactant productivity, *Y*_2_, which indicates the effectiveness of the microbe to produce high concentrations of biosurfactant in the shortest period of time, proved that the quadratic model was highly significant (*p* = 0.001) in which there was only a 0.01% probability that such a large model F value could be generated due to noise (Appendix A). The significant square terms (A: PFAD, D: sodium-EDTA and E: MgSO_4_·7H_2_O) were identified with the *ρ*-value less than 0.10 and other terms only exist as linear predictor terms. In addition, only one interactive predictor term which was PFAD*KH_2_PO_4_ (A*B) that was significant to the model (*p* = 0.0474) (Figure 2b).

The equation to represent the model for maximum biosurfactant productivity by *Pseudomonas* sp. LM19 is described by Equation (2). The value of R^2^ for this model is 0.8097, demonstrated that 80.97% of the variability in the *Y*_2_ response could be clarified by Equation (2). A difference of less than 0.2 between the value of ‘R^2^ predictive’ (0.6877) and ‘R^2^ adjusted’ (0.7680) suggests that this model was fitted with further supporting data of non-significant value for lack of its fit test. Other than that, a relatively low coefficient of variation (CV = 24.81%) indicated the precision and reliability of the model.
(2)$$\begin{array}{l} Maximum\;biosurfactant\;productivity,\;Y_{2}\\ =0.33+0.03\;PFAD-0.025\;sodium•EDTA+0.013\;MgSO_{4}\cdot 7H_{2}O-0.017\;PFAD*KH_{2}PO_{4}\\ -0.044\;PFAD*PFAD-0.021KH_{2}PO_{4}*KH_{2}PO_{4}-0.043\;yeast\;extract*yeast\;extract\\ -0.018\;sodium•EDTA*sodium•EDTA-0.044\;MgSO_{4}\cdot 7H_{2}O*MgSO_{4}\cdot 7H_{2}O \end{array}$$

For this study, PFAD was the sole carbon source chosen for biosurfactant production. Equation (2) shows that PFAD contributed positively when present in high concentrations due to the fact that it contains a bunch of fatty acids in their composition, which *Pseudomonas* sp. LM19 could utilize for energy sources or proceed to hydrolyze them to become part of the hydrophobic tails biosurfactant structure. For instance, the presence of carboxylic acid group was detected in the biosurfactant produced by *P. aeruginosa* TMN when utilizing glycerol and glucose as substrates [38]. The second quadratic term (sodium-EDTA) in Equation (2) produces negative impacts to biosurfactant production when present in a high concentration as similar to cell growth. Sodium-EDTA is known as an inhibitor for biosurfactant production when presented in a high concentration in the culture medium but become a good chelating agent for better utilization of a metal ion by microorganisms when present in the proper amount [29]. The existence of MgSO_4_·7H_2_O as a single term in high concentration gave a positive effect on biosurfactant productivity. In particular, an Mg^2+^ ion is required for the cell to activate the energy by ATP. This interaction will cause the increment of the Mg^2+^ ion usage by cells when higher metabolic activity was identified just like in the case for the production of secondary metabolites, biosurfactant [18]. This can be proved when *Halobacterium salinarium* obtained the double amount of biosurfactant (E24 = 68%) when using MgSO_4_ in the production as compared to other metal ions with the same concentration [39]. However, the quadratic term of it produces a negative effect towards the maximum biosurfactant productivity, hinting at an excessive amount of MgSO_4_·7H_2_O that could produce lethal effects. Figure 2b demonstrates the response surface and contour plots generated using Equation (2) to present the interactive effect between KH_2_PO_4_ and PFAD towards biosurfactant productivity. Each figure demonstrates the effect of changing those two variables (KH_2_PO_4_ and PFAD) on biosurfactant productivity during fermentation by *Pseudomonas* sp. LM19. The model prediction of maximum biosurfactant productivity can reach up to 0.347 g/L/day in optimal medium composition.

#### 2.5.3. Validation of the Experiment

In order to validate the optimum culture medium as predicted by the RSM model, *Pseudomonas* sp. LM19 was cultivated in flasks in the same conditions described previously, using a production medium with the following composition: 1.148% (*v*/*v*) PFAD; 4.054 g/L KH_2_PO_4_; 1.30 g/L yeast extract; 0.023 g/L sodium-EDTA; 1.057 g/L MgSO_4_·7H_2_O; 0.75 g/L K_2_HPO_4_; 0.20 g/L CaCl_2_·2H_2_O; 0.080 g/L FeCl_3_·6H_2_O. The cell concentration produced from these conditions was 8.53 × 10^9^ CFU/mL after seven days of growth, which is almost similar to the cell concentration value predicted by the model (8.396 × 10^9^ CFU/mL). For maximum biosurfactant productivity, the highest achievement was 0.3463 g/L/day (2.424 g/L on the 7th day of fermentation), which is comparable to the value predicted from the model (0.347 g/L/day). Therefore, it can validate that both models competently conform to the experimental data and apparently interpret the effect of the production medium composition on cell growth and maximum biosurfactant productivity.

### 2.6. Time Course Data for Biosurfactant Production by Pseudomonas sp. LM19 in Optimized Medium Conditions

The optimized culture medium was used to produce biosurfactant once again in shake flasks to obtain full profiling of cell growth, biosurfactant production and PFAD consumption. All of the original environmental conditions were retained and the sampling was done every two days. The fermentation profiles are shown in Figure 3. The cells grew rapidly with no apparent lag phase from the first day (µ_max_ = 1.11) and slowly entering the stationary phase at around 48 h. A maximum cell concentration (8.58 × 10^9^ CFU/mL) was achieved after six days of fermentation. Production of biosurfactant increased significantly cell had entered the stationary phase and reached maximum concentration (1.84 g/L) on the 6th day of fermentation, suggesting the biosurfactant production to be non-growth associated. In most of the previous work, biosurfactant production by *Pseudomonas* sp. were non-growth associated as biosurfactants are well known to be secondary metabolites [40,41]. PFAD in culture broth decreased rapidly for the first two days of fermentation but seemed relatively unconsumed (0.32% *v*/*v*) when entering 6th day onwards, suggesting that a high amount of PFAD were being utilized in this period to produce biosurfactant and to maintain high cell production. 

Previous work in utilizing PFAD as a carbon source to produce biosurfactant from *Pseudomonas* sp. had employed higher initial concentration (20 to 100 g/L) compared to this work [13]. Nonetheless, maximum biosurfactant (BS_max_) gained was much lower, ranging from 0.39 to 0.43 g/L, suggesting that higher concentrations of PFAD used could negatively affect the biosurfactant production. In addition, the productivity of biosurfactant (P_BS_) achieved in this work was higher (0.010 g/L/h) compared to previous work (0.004 g/L/h) showing higher capability of biosurfactant production by strain *Pseudomonas* sp. LM19. Furthermore, this study was done on a lab scale while the previous study conducted their production in a 5 L bioreactor, which makes this study much more effective compared to previous study and showing the potential for scaling up. In some other cases, *P. aeruginosa* PAO1 showed higher productivity (0.43 g/L/h) when utilizing sunflower oil compared to PFAD [42] due to the difference of production scale used by sunflower oil which uses 30 L bioreactors compared to PFAD (5 L bioreactor). In addition, the utilization of palm oil as a substrate for *P. aeruginosa* A41 to produce rhamnolipid showed a higher level of maximum biosurfactant achieved and productivity value (BS_max_ = 2.91 g/L, P_BS_ = 0.017 g/L/h) compared to that observed in this study [43]. Higher FFA content present in PFAD at the same concentration to that palm oil could lead to the excessive fatty acid in culture broth, which might not be favourable to the growth of the microorganisms unless set at an optimum level [44]. On the other hand, this study achieved a higher product yield (Y*_P_*_/*S*_ = 0.27 g/g) compared to the study by Moya et al. [45] which utilized olive mill waste by *P. aeruginosa* PAO1 to produce surfactin even in different concentrations of substrate (20, 50 and 100 g/L), which corresponded to 0.0130, 0.0180 and 0.0580 g/g, respectively. The results demonstrated that the production yield of biosurfactant was improved when the concentration of olive mill waste used was increased, which contradicts the findings of our study. On the other hand, the product yield reported by Nazren Radzuan [13], which also utilized PFAD as substrate, was lower (Y*_P_*_/*S*_ = 0.193 g/g) as compared to the product yield obtained in this study in which both studies were using PFAD as substrate. This indicates that the production of biosurfactant is strain and substrate dependent. However, *P. aeruginosa* PA1 achieved almost similar Y*_P_*_/*S*_ (0.103) to this study while utilizing hydrolyzed glycerine as a carbon source in 0.25 L flasks [46]. 

### 2.7. Identification of Biosurfactants Class and Congeners

The identification of the structural constituents of the purified biosurfactant was done by using HPLC-MS. According to the HPLC-MS data acquired (Table 5), nine rhamnolipid congeners were detected, with 57.35% of dirhamnolipid and the rest were monorhamnolipid. *P. aeruginosa* LM19 yielded a major component with a predominant peak at *m*/*z* 649.54 and a second major component at *m*/*z* 503.42, which corresponds to the deprotonated molecules [M-H]^−^ of the dirhamnolipid [Rha-Rha-C_10_-C_10_] and the monorhamnolipid [Rha-C_10_-C_10_], respectively. These types of rhamnolipid congeners are usually being produced by *P. aeruginosa* as detailed in the studies by several investigators [47,48,49]. Other rhamnolipid congeners with the minor amounts can be identified with the presence of a peak at *m*/*z* 677.58 and 475.41 which represented [Rha-Rha-C_12_-C_10_/Rha-Rha-C_10_-C_12_] and [Rha-C_10_-C_8_/Rha-C_8_-C_10_]. It can be concluded that the biosurfactant produced by *P. aeruginosa* LM19 strain, utilizing PFAD as sole carbon source comprises of both mixtures between mono and dirhamnolipids. 

According to the previous literature, different carbon sources used during the fermentation could influence the characteristics and the number or amount of rhamnolipid congeners produced at the end of the fermentation process. For example, Kristoffersen et al. [51] identified the presence of six types of rhamnolipids congeners when an arctic marine bacterium from the *P. fluorescence* group was cultivated with mannitol as a carbon source. In comparison to Thio et al. [50] opting for palm kernel fatty acid distillate (PKFAD) as a sole carbon source, their results also demonstrated that the same major rhamnolipid congeners were produced at the end of fermentation by virtue of PKFAD having almost the same composition as PFAD but with a lesser amount of FFA. However, a different strain of *Pseudomonas* sp. also could yield different composition, abundance and concentration of congeners in rhamnolipids produced ranging from C_8_ to C_12_ when using sunflower oil as a carbon source [52]. In terms of survival, monorhamnolipids mainly act as wetting agents, which are responsible for enhancing the transport by reducing the surface tension, and does not play a major role in modulating swarming motility of the microorganisms itself [53], while dirhamnolipid serves as an attractant, which is identified by bacterial swarmers. Other properties of monorhamnolipid are having less solubility, stronger surface sorption, and stronger cationic metals binding compared to dirhamnolipid [54]. On the other hand, dirhamnolipid possesses greater emulsion stability and foam formation due to the presence of two molecules of rhamnose [55]. These properties are important to differentiate their role in different industries. For example, dirhamnolipid showed dose-dependent mortality against aphids *(Myzus persicae)*, producing about 50% mortality at 40 μg/mL and 100% mortality at 100 μg/mL proving considerable insecticidal activity in the agricultural sector [56]. On the other hand, monorhamnolipid produced from *Candida tropicalis* exhibited potential application in the bioremediation of hydrocarbons [57]. The result showed that the degradation of hexadecane was improved when optimum concentration (19 mg/L) of monorhamnolipid was used. In addition, the presence of monorhamnolipid altered the cell surface properties demonstrated the reason for the enhanced biodegradation of hexadecane by the yeast.

## 3. Materials and Methods

### 3.1. Microorganism Preservation and Sub-Culturing Process

The bacterium strain *Pseudomonas* sp. LM19 used in the present study was isolated from a chicken processing factory in Pedas, Negeri Sembilan, Malaysia. A BLAST (Basic Local Alignment Search Tool) alignment of the 16S rRNA with the GenBank database of the National Center for Biotechnology Information (NCBI) (Bethesda, Maryland) indicated a 99% of sequence similarity with members of the genus *Pseudomonas*, such as *Pseudomonas aeruginosa* strain DQ2. *Pseudomonas* sp. LM19 was deposited as strain UPMC 1111 in the Microbial Culture Collection Unit (UNiCC), Bioscience Institute, Universiti Putra Malaysia, Serdang, Malaysia. It was maintained in 40% glycerol stock stored in −80 °C prior to use. The culture was streaked onto LB agar and incubated for 24 h in 30 °C before being employed as inoculum in the following fermentation process.

### 3.2. Palm Fatty ACID Distillate (PFAD) 

PFAD was kindly provided by North Emerald Pte. Ltd., Klang, Selangor, Malaysia and its FFA composition was determined by gas chromatography-mass spectrometry (GC-MS) (Agilent, Santa Clara, FL, USA). The preparation of fatty acid methyl esters (FAME) was done through esterification and hydrolysis process using a standard American Oil Chemists′ Society AOCS method AOCS Ce2-66, 2009 [58] with modifications. The PFAD was melted at 60 °C overnight before esterification process took place. A 1 mL of toluene was added into 50 µL of PFAD in the test tube before mixed with 2 mL of 2% (*v*/*v*) of acidified methanol. The test tube was connected to a reflux condenser and was heated to 80 °C for 2 h. After the esterification process, the product was allowed to cool before 5 mL of 5% (*w*/*v*) sodium chloride and 3 mL of hexane was added to extract the FAME. Two distinct layers were formed after the mixture was centrifuged at 2500 rpm for 5 min. The extraction process was repeated twice and the top layer (organic layer) was pooled and 5 mL of 2% (*w*/*v*) of potassium bicarbonate was added. The mixture was then centrifuged and dried with nitrogen gas. The produced FAME was analysed by GC–MS (7890A GC with model 5975C mass selective detector), the FAME was dissolved in 1 mL of hexane and a capillary column DB-WAXetr (length: 30 m, ID: 250 µm and film thickness: 0.25 µm; Agilent). A 2 µL of the sample was then injected into the GC–MS injector port. The GC oven was programmed with an increasing starting temperature from 50 °C for 1 min to 200 °C with 25 °C/min and continues to 3 °C/min to 230 °C for 5 min.

### 3.3. Inoculum Preparation and Fermentation Procedure

One loopful of culture *Pseudomonas* sp. LM19 from LB agar was inoculated into 100 mL of LB broth in 250 mL Erlenmeyer flask and incubated in a rotary shaker at 180 rpm in 30 °C for 22 h or until the OD_600_ reached 1.5 This was used as inoculum with 1% (*v*/*v*) inoculum size together with 1% (*v*/*v*) PFAD concentration as a carbon source in different production media. Fermentation was done in a 250 mL Erlenmeyer flask with 100 mL total working volume for 10 days. The sampling was done at every 24 h to determine the cell concentration (OD_600_ and Colony Forming Unit) and biosurfactant concentration (Emulsification index, E24 and orcinol assay). Experiments were conducted in triplicate.

### 3.4. Selection of Production Medium

The selection of the media used in this study satisfied the following conditions; (i) biosurfactant producer involved was *Pseudomonas* sp., (ii) a wide range of substrates was covered, and (iii) operated in lab scale. Five different production media tested were LB (Luria Bertani); 10 g/L peptone, 5 g/L yeast extract, 10 g/L NaCl, BH (Bushnell–Haas); 1 g/L KH_2_PO_4_, 1 g/L of K_2_HPO_4_, 1 g/L NH_4_NO_3_, 0.2 g/L MgSO_4_·7H_2_O, 0.2 g/L CaCl_2_·2H_2_O, 0.05 g/L FeCl_3_·6H_2_O [59], Modified BH (Bushnell–Haas); original composition of BH with addition of 0.125 g/L sodium-EDTA, MS (Mineral Salt); 4.08 g/L KH_2_PO_4_, 5.68 g/L Na_2_PHO_4_, 4.0 g/L NH_4_NO_3_, 0.0015 g/L sodium-EDTA, 0.007 g/L of CaCl_2_, 0.2 g/L MgSO_4_·7H_2_O, 0.0006 g/L FeSO_4_·7H_2_O [34] and, finally, PPAS (Phosphate-limited Proteose-peptone Ammonium Salt); 0.07 g/L NH_4_Cl, 10 g/L glycine, 1.49 g/L KCl, 0.2 g/L MgSO_4_ and 14.54 g/L Tris-HCl [60], to identify the most suitable production medium in producing a maximum amount of biosurfactant. The pH of media was adjusted to 7 ± 0.02 and sterilized by autoclaving them at 121 °C for 15 min. The biosurfactant production was analysed using orcinol test, emulsification index (E24) and surface tension value while the cell growth was measured using Colony Forming Unit (CFU/mL) and optical density (OD_600_). 

### 3.5. Optimization Procedures of Parameters 

#### 3.5.1. One-Factor-at-Time (OFAT)

The OFAT was used to investigate the range of main process parameters involved in biosurfactant production. Process parameters were optimized by using modified BH as the basal medium in a series of experiments to obtain a higher concentration of biosurfactant. The observed variables were inoculum sizes, PFAD concentrations, a different type of nitrogen sources used at constant nitrogen content (0.35 g/L), and later the concentration of the chosen nitrogen source. The inoculum sizes used ranged from 1%, 2%, 4%, 6% and 8% (*v*/*v*) while PFAD concentration tested from 0.5%, 1%, 2%, 4% and 6% (*v*/*v*). Nitrogen sources tested were ammonium nitrate (NH_4_NO_3_), ammonium sulphate [(NH_4_)_2_SO_4_], sodium nitrate (NaNO_3_), yeast extract, soytone and fish meal. Later, the concentration of chosen nitrogen source that was experimented with ranged between 0.5, 1.5, 3, 4.5 and 6 g/L.

#### 3.5.2. Factorial Screening by Plackett–Burman Design (PBD)

PBD feature in Design Expert software (version 10, Stat-Ease Inc., Minneapolis, MN, USA) was used to screen seven components in modified BH medium in 13 randomized experimental runs (Table 2). The PBD was used to identify the significant medium components of selected production medium according to Section 2.4 which contributed the most to both biosurfactant production and cell growth. The components included A: KH_2_PO_4_, B: K_2_HPO_4_, C: MgSO_4_·7H_2_O, D: CaCl·2H_2_O, E: yeast extract F: FeCl_3_·6H_2_O, G: sodium-EDTA. Each component was tested at two different levels; +1 and –1 (Table 6), with biosurfactant productivity (expressed as the maximum concentration of biosurfactant produced divided by the corresponding incubation time in days), *Y*_1_ and Colony Forming Unit (CFU), *Y*_2_ served as the response variables. Data were analysed using the same statistical software that generated the design. Significant medium components were selected for further optimization by RSM optimization.

#### 3.5.3. Optimization via Response Surface Methodology (RSM)

A CCD was adopted to further optimize the significant medium components identified from PBD. The test variables used were A: PFAD, B: KH_2_PO_4_, C: yeast extract, D: sodium-EDTA and E: MgSO_4_·7H_2_O. A total of 53 experimental runs were performed in three different blocks (Table 7). 

The second order polynomial coefficients were analysed and calculated using Design Expert statistical software. The behaviour of the system can be explained in Equation (3) below:(3)$$Y_{i}{\;=\;B}_{0}\;+\;\sum_{i\;=\;1}^{n}B_{i}\chi_{i}\;+\;\sum_{i<j}^{n}B_{ij}\chi_{i}\chi_{j}\;+\;\sum_{j\;=\;1}^{n}B_{ii}\chi_{i}^{2},$$where $Y_{i}$ is the predicted response and *n* represents the variables involved in this study while $\chi_{i}\chi_{j}$ are the input variables affecting the response variable, *Y*; $B_{0}$ is the offset term; $B_{i}$ is the *i*th linear coefficient; $B_{ii}$ is the *i*th quadratic coefficient and $B_{ij}$ is the *ij*th interaction coefficient. The response variables were biosurfactant productivity, *Y*_1_ (g/L/day) and cell growth, *Y*_2_ (×10^9^ CFU/mL), respectively. A study to confirm the validity of the optimization process was done by running additional *Pseudomonas* sp. LM19 fermentation using the best predicted medium composition of PFAD, KH_2_PO_4_, yeast extract, MgSO_4_·7H_2_O and sodium-EDTA. 

### 3.6. Identification of Biosurfactant Class and Congeners Produced

The production broth from the optimized medium was first extracted according to the method by Patowary et al. [61] with slight modifications. The cell-free was acidified to pH 2 using 1 M hydrochloric acid (HCl) and stored in 4 °C for 15 to 20 h. Ethyl acetate with ratio 1:1 was added to the acidified supernatant and was vortexed for one minute. The mixture was then left stationary for layer separation. The ethyl acetate layer was collected and transferred to a vacuum evaporator (DyNA) to be concentrated at 40 °C under negative pressure. The honey-like viscous liquid produced after the recovery process (10 mg) was resuspended in 1 mL of methane and it was subjected to HPLC analysis to determine their biosurfactants class and congeners. The Liquid Chromatography (LC) flow rate was set at 1 mL/min and the injection volume was 10 µL/min. For the mobile phase, an acetonitrile-water (0.1% formic acid) gradient was used starting with 40% of acetonitrile and 60% of water for 5 min. Then, acetonitrile concentration was raised from 40% to 100% (with water decreasing in concentration simultaneously) across a period of 4 min 30 s. The gradient was maintained at 100% acetonitrile for 1 min and then returned to initial conditions over a period of 2 min. Mass spectrometry (MS) was performed with a QDa mass detector, equipped with an evaporative light scattering detector (ELSD). A range of *m*/*z* 100 to 1000 data was scanned and obtained (modified from Haba et al. [62]). 

### 3.7. Analytical Method

#### 3.7.1. Cell Density Measurement

The cell population density was measured via spectrophotometry as well as colony forming unit (CFU). The pellet settled after centrifugation (10,000 rpm, 10 min) of the medium was washed and re-suspended twice in 0.85% of NaCl solution and their density was measured at 600 nm (OD_600_) [63]. CFU was measured by plating 100 µL of fermentation broth diluted to the factor of 10^6^, 10^7^ and 10^8^ onto a nutrient agar plate and incubated at 30 °C for 24 h. The CFU/mL was calculated based on Equation (4):(4)$$\mathrm{CFU}/\mathrm{mL}\;=\;\frac{\mathrm{No}.\;\mathrm{of}\;\mathrm{colonies}\;\mathrm{counted}}{\mathrm{The}\;\mathrm{volume}\;\mathrm{of}\;\mathrm{sample}\;\mathrm{plated}\;(\mathrm{mL})}\;\times \;\mathrm{Dilution}\;\mathrm{factor}\;$$

#### 3.7.2. Emulsification Index (E24) 

The emulsification index (E24) was determined whereby 2 mL of cooking oil (Buruh refined cooking oil) was added into an equal volume of cell-free supernatant followed by vortexing them for two minutes [64]. They were allowed to stand for 24 h in room temperature and E24 index can be determined by Equation (5): (5)$$E24\;\mathrm{index}\;\left(\%\right)\;=\;\frac{\mathrm{Height}\;\mathrm{of}\;\mathrm{emulsified}\;\mathrm{layer}\;(\mathrm{cm})}{\mathrm{The}\;\mathrm{total}\;\mathrm{height}\;\mathrm{of}\;\mathrm{the}\;\mathrm{liquid}\;\mathrm{column}\;(\mathrm{cm})}\;\times \;100\%\;$$

#### 3.7.3. Surface Tension 

The surface tension of the culture supernatants was measured with a digital surface tensiometer (Kibron AquaPi, Helsinki, Finland) using a rod-in-free-surface (RIFS) technique [65]. For calibration, the surface tension of deionized water was first measured and, after each reading, the rod was heated until red hot under the flame before being used for each new sample.

#### 3.7.4. Biosurfactant Extraction and Quantification

Biosurfactant was recovered according to the method described by Nazren Radzuan et al. [13] with slight modification. The supernatant was mixed with n-hexane in the ratio of 2:1 to remove excess PFAD from culture broth and centrifuged at 10,000 rpm for 10 min. The cell-free supernatant was acidified with 1 M hydrochloric acid (HCl) until its pH reached 2 and then the mixture was stored in 4 °C, overnight for complete precipitation. The supernatant containing precipitate was then mixed with the same volume of ethyl acetate and allowed to stand until two layers were formed. The ethyl acetate layer was pooled and concentrated under vacuum pressure using vacuum evaporator (DyNA) to obtain the crude biosurfactant at 40 °C and top-up to original volume by using distilled water. Crude biosurfactant was quantified using orcinol assay [66]. For every 100 µL of extracted biosurfactant, 900 µL of solution containing 0.19% (*v*/*v*) of orcinol in 53% (*w*/*v*) of H_2_SO_4_ was added together. The mixture was incubated at 80 °C for 30 min and cooled at room temperature before measuring the absorbance reading at a wavelength of 421 nm (OD_421_). The blank used was distilled water with the addition of orcinol reagent. Rhamnolipid concentrations were calculated from a standard curve prepared with L-rhamnose and expressed as rhamnose equivalents (RE) (g/L). 

#### 3.7.5. Determination of Residual PFAD

This method was modified from Müller et al. [42] by using 1 mL of fermentation broth mixed with 0.5 mL of n-hexane in the centrifuge tube. The mixture was then vortexed for 1 min and centrifuged at 10,000 rpm for 10 min. Following a distinct two-layer formation, the hexane (upper) layer which contained remaining PFAD was collected and transferred into a pre-weighed centrifuge tube. The collected hexane phase was then air-dried for 3 days and stored in a desiccator overnight before weighing them on measuring balance until it registered constant reading. The remaining PFAD was calculated based on Equation (6): (6)$$\mathrm{PFAD}\;\left(\mathrm{mL}/L\right)\;=\;\frac{\mathrm{Tube}\;\mathrm{with}\;\mathrm{PFAD}\;\mathrm{layer}\;\left(g\right)-\mathrm{Empty}\;\mathrm{tube}\left(g\right)}{\mathrm{The}\;\mathrm{volume}\;\mathrm{of}\;\mathrm{the}\;\mathrm{sample}\;(L)}\times \frac{1}{\mathrm{The}\;\mathrm{density}\;\mathrm{of}\;\mathrm{PFAD}\;(g/\mathrm{mL})}$$

### 3.8. Statistical Analysis

All experiments were conducted at least in triplicate and the results represented are the means with the standard deviation (SD) of the independent runs. A one-way ANOVA test using the least significant difference (LSD) was conducted using SPSS version 16.0 (SPSS Inc., Chicago, IL, USA) at 95% confidence to determine the significance of every data presented in this work. On the other hand, statistical analysis of models’ fitness obtained from PBD and RSM was performed by ANOVA, which is readily available in Design Expert. 

## 4. Conclusions

The optimal medium composition for the production of biosurfactant in batch shake flask fermentation was developed resulting in maximum biosurfactant productivity with optimum cell density. Sequential optimization using OFAT and CCD increased the biosurfactant productivity by 3.33-fold compared to the non-optimized production medium. Full-time course data of cell generation, biosurfactant production and substrate (PFAD) consumption by microbes in optimized medium were established and some comparisons of kinetic parameters with the previous work to show the kinetic improvement in the process. In addition to that, biosurfactant produced in this work was comprised of 42.19% mono and the rest were dirhamnolipid. In general, this work demonstrated a good strategy to develop an optimized production medium comprised of both macro and micronutrients by using both traditional and statistical approaches. Furthermore, this study demonstrated the feasibility of using PFAD in an optimized medium for maximizing the biosurfactant production, which may add value to the downstream process in the palm oil industry, in regard to economic and environmental value. On the other hand, the identification of biosurfactant class and congeners specified their application in different industries for future use.

## Figures and Tables

**Figure 1 molecules-24-02613-f001:**
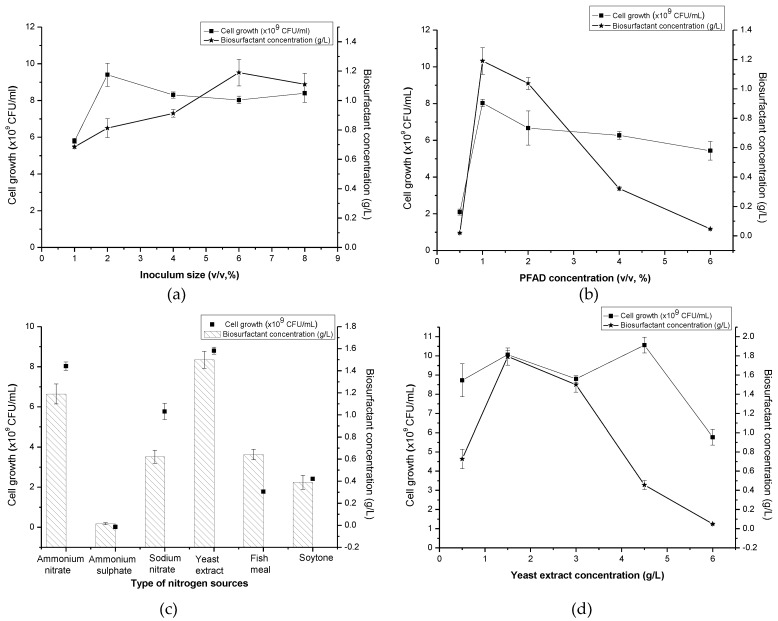
Effect of (**a**) inoculum size; (**b**) PFAD concentration; (**c**) type of nitrogen sources; and (**d**) yeast extract concentration on cell growth and biosurfactant concentration by *Pseudomonas* sp. LM19.

**Figure 2 molecules-24-02613-f002:**
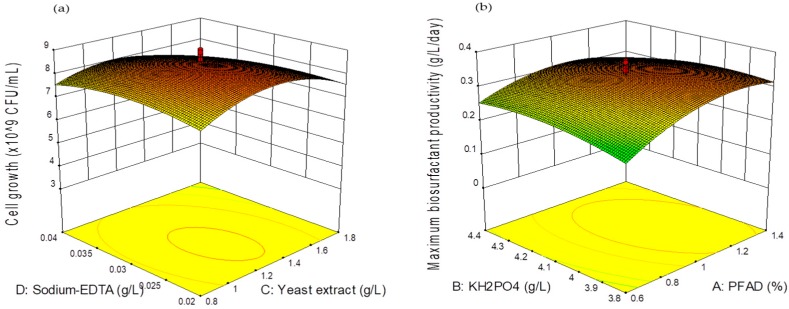
Response surface plot showing the interactive effect of (**a**) yeast extract and sodium-EDTA concentration on the cell growth and (**b**) KH_2_PO_4_ and PFAD on maximum biosurfactant productivity.

**Figure 3 molecules-24-02613-f003:**
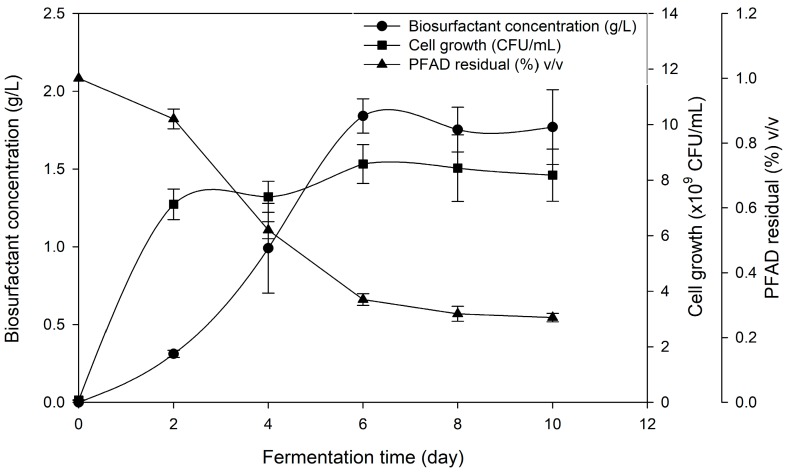
The profiles of microbial growth, biosurfactant production and PFAD residual in the broth during fermentation performed by *Pseudomonas* sp. LM19 grown in the optimized medium.

**Table 1 molecules-24-02613-t001:** Free fatty acids composition in PFAD.

Name	Formula	Composition (%)
Palmitic acid	C_16_H_32_O_2_	50.30
Oleic acid	C_18_H_34_O_2_	28.09
Lauric acid	C_12_H_24_O_2_	6.67
Stearic acid	C_18_H_36_O_2_	6.07
Linoleic acid	C_18_H_32_O_2_	4.38
Myristic acid	C_14_H_28_O_2_	3.77
Gamolenic acid	C_18_H_30_O_2_	0.37
Capric acid	C_10_H_20_O_2_	0.36
Total		100

**Table 2 molecules-24-02613-t002:** Effect of different production media used towards cell growth and biosurfactants production by *Pseudomonas* sp. LM19.

Analysis/Production Media (Incubation Time)	Cell Growth		Biosurfactants (BS)		
	Colony Forming unit (×10^8^ CFU/mL)	Maximum OD_600_	BS_max_ (g/L)	Emulsification index, E24_max_ (%)	Surface Tension_max_ (mN/m)
LB (Day 6)	170 ± 3 ^a^	2.211 ± 0.066 ^a^	0.19 ± 0.01 ^a^	52.56 ± 2.22 ^a^	41.28 ± 0.04 ^a^
BH (Day 9)	95 ± 4 ^b^	1.868 ± 0.166 ^b^	0.05 ± 0.01^b^	46.15 ± 3.85 ^a^	52.01 ± 0.04 ^b^
Modified BH (Day 9)	58 ± 1 ^c^	1.539 ± 0.056 ^c^	0.68 ± 0.01 ^c^	59.62 ± 2.72 ^b^	33.11 ± 0.02 ^c^
MS (Day 7)	11 ± 1 ^d^	1.301 ± 0.050 ^d^	0.14 ± 0.02 ^d^	53.13 ± 2.85 ^b^	43.52 ± 0.03 ^d^
PPAS (Day 5)	2 ± 0 ^e^	0.027 ± 0.020 ^e^	0.04 ± 0.02 ^b^	0.00 ± 0.00 ^c^	54.05 ± 0.02 ^b^

* Value is mean ± SD (*n* = 3); different alphabet superscript indicates significant differences (*p* < 0.05) in both cell growth and biosurfactant production in the medium.

**Table 3 molecules-24-02613-t003:** Plackett–Burman experimental design for seven variables and the corresponding experimental response (cell growth and maximum biosurfactant productivity).

Run	Variables Code							Responses’ Variables	
	A	B	C	D	E	F	G	Cell Growth, *Y*_1_ (×10^9^ CFU/mL)	Maximum Biosurfactant Productivity, *Y*_2_ (g/L/day)
1	0.2	7	1	0.01	4.5	0.08	0.3	8.1	0.0630
2	4	0.75	1	0.2	4.5	0.05	0.3	11.9	0.1690
3	0.2	7	1	0.2	0.5	0.05	0.03	7.8	0.0980
4	0.2	0.75	0.04	0.2	4.5	0.05	0.3	8.3	0.0067
5	0.2	0.75	1	0.01	4.5	0.08	0.03	12.0	0.0390
6	4	0.75	0.04	0.01	0.5	0.08	0.3	7.6	0.0922
7	4	0.75	1	0.2	0.5	0.08	0.03	9.0	0.2722
8	2.1	3.875	0.52	0.105	2.5	0.065	0.165	8.7	0.1389
9	4	7	1	0.01	0.5	0.05	0.3	7.3	0.2934
10	4	7	0.04	0.2	4.5	0.08	0.03	14.2	0.0076
11	4	7	0.04	0.01	4.5	0.05	0.03	14.0	0.0051
12	0.2	7	0.04	0.2	0.5	0.08	0.3	7.9	0.0449
13	0.2	0.75	0.04	0.01	0.5	0.05	0.03	7.1	0.0148

Notes: A: KH_2_PO_4_, B: K_2_HPO_4_, C: MgSO_4_·7H_2_O, D: CaCl·2H_2_O, E: yeast extract F: FeCl_3_·6H_2_O, G: sodium-EDTA.

**Table 4 molecules-24-02613-t004:** Actual factor levels corresponding to coded factor levels for the CCD of the experimental response surface optimization showing cell growth and maximum biosurfactant productivity.

Block	Run	Variables Codes					Responses	
		A	B	C	D	E	Cell Growth, *Y*_1_ (×10^9^ CFU/mL)	Maximum Biosurfactant Productivity, *Y*_2_ (g/L/day)
1	1	1.40	4.40	0.80	0.02	0.60	5.4	0.1981
	2	0.60	4.40	0.80	0.02	1.40	5.6	0.2985
	3	1.00	4.10	1.30	0.03	1.00	8.9	0.3436
	4	1.40	3.80	1.80	0.02	0.60	5.6	0.1519
	5	0.60	4.40	1.80	0.02	0.60	6.0	0.1664
	6	1.40	3.80	0.80	0.02	1.40	6.1	0.2643
	7	1.40	4.40	0.80	0.04	1.40	5.1	0.1825
	8	0.60	3.80	1.80	0.02	1.40	5.6	0.1468
	9	1.40	3.80	1.80	0.04	1.40	5.1	0.1684
	10	1.00	4.10	1.30	0.03	1.00	8.8	0.3319
	11	1.00	4.10	1.30	0.03	1.00	8.9	0.3538
	12	1.00	4.10	1.30	0.03	1.00	8.4	0.3745
	13	0.60	3.80	0.80	0.02	0.60	5.8	0.1267
	14	0.60	4.40	0.80	0.04	0.60	5.8	0.1090
	15	0.60	4.40	1.80	0.04	1.40	4.5	0.0607
	16	0.60	3.80	0.80	0.04	1.40	5.6	0.0679
	17	1.40	4.40	1.80	0.02	1.40	5.4	0.2179
	18	1.40	3.80	0.80	0.04	0.60	6.8	0.1805
	19	0.60	3.80	1.80	0.04	0.60	4.8	0.0227
	20	1.40	4.40	1.80	0.04	0.60	4.7	0.142
2	21	1.40	4.40	0.80	0.04	0.60	6.3	0.1507
	22	0.60	3.80	1.80	0.04	1.40	5.0	0.1709
	23	1.40	3.80	1.80	0.02	1.40	5.5	0.1803
	24	0.60	3.80	1.80	0.02	0.60	6.3	0.0815
	25	1.40	3.80	0.80	0.04	1.40	5.2	0.1225
	26	0.60	3.80	0.80	0.04	0.60	6.6	0.1443
	27	1.00	4.10	1.30	0.03	1.00	9.0	0.3473
	28	0.60	4.40	0.80	0.02	0.60	6.6	0.1223
	29	1.00	4.10	1.30	0.03	1.00	8.9	0.3699
	30	1.40	3.80	0.80	0.02	0.60	6.7	0.3464
	31	1.00	4.10	1.30	0.03	1.00	8.5	0.3705
	32	0.60	4.40	1.80	0.02	1.40	4.6	0.1738
	33	0.60	4.40	0.80	0.04	1.40	5.2	0.1246
	34	1.40	4.40	1.80	0.04	1.40	5.0	0.1370
	35	1.00	4.10	1.30	0.03	1.00	8.6	0.3008
	36	0.60	4.40	1.80	0.04	0.60	5.2	0.1260
	37	1.40	3.80	1.80	0.04	0.60	5.1	0.2424
	38	1.40	4.40	1.80	0.02	0.60	6.8	0.1393
	39	1.40	4.40	0.80	0.02	1.40	5.2	0.1982
	40	0.60	3.80	0.80	0.02	1.40	4.2	0.1570
3	41	1.00	4.10	1.30	0.03	0.05	5.6	0.0183
	42	1.00	4.10	1.30	0.03	1.00	7.0	0.3264
	43	1.00	4.10	1.30	0.03	1.00	8.2	0.3316
	44	1.00	4.10	1.30	0.01	1.00	8.0	0.2902
	45	1.00	4.10	1.30	0.05	1.00	6.1	0.1847
	46	1.00	4.10	0.11	0.03	1.00	4.7	0.0651
	47	1.00	4.10	1.30	0.03	1.95	4.2	0.1553
	48	0.05	4.10	1.30	0.03	1.00	5.9	0.0089
	49	1.00	4.10	2.49	0.03	1.00	3.6	0.1201
	50	1.00	4.10	1.30	0.03	1.00	8.9	0.3661
	51	1.00	4.81	1.30	0.03	1.00	8.3	0.2106
	52	1.00	3.39	1.30	0.03	1.00	6.3	0.2318
	53	1.95	4.10	1.30	0.03	1.00	6.2	0.1664

Notes: A: PFAD, B: KH_2_PO_4_, C: yeast extract, D: sodium-EDTA, E: MgSO_4_·7H_2_O.

**Table 5 molecules-24-02613-t005:** Rhamnolipid congeners detected by HPLC-MS in crude biosurfactant produced.

No.	Rhamnolipids Congeners	Pseudomolecular Ion (*m*/*z*)	Relative Abundance (%)	
			This Work	[50]
1	RRC_10_C_8_/RRC_8_C_10_	621.49	0.21	-
2	RRC_10_C_14:1_	696.12	0.46	-
3	RC_10_C_8_/RC_8_C_10_	475.41	1.30	-
4	RRC_10_C_10_	649.54	54.18	45.98
5	RRC_12:1_C_10_/RRC_10_C_12:1_	675.56	0.14	8.10
6	RRC_12:1_C_10_/RRC_10_C_12:1_	675.54	0.50	-
7	RC_10_C_10_	503.42	39.94	33.22
8	RRC_12_-C_10_/RRC_10_C_12_	677.58	1.93	3.34
9	RC_10_C_12:1_/RC_12:1_C_10_	529.41	0.41	8.07
10	RRC_10_C_12:1_/RRC_12:1_C_10_	529.44	0.39	-
11	RC_10_C_12_/RC_12_C_10_	531.46	0.53	1.29

**Table 6 molecules-24-02613-t006:** Plackett–Burman Design with low (−) and high (+) levels together with one center point array (0) for seven nutritional factors in modified BH medium.

Levels	Independent Variables Codes (Nutritional Factors) (g/L)						
	A	B	C	D	E	F	G
+	4	7	1	0.2	4.5	0.08	0.3
0	2.1	3.875	0.52	0.105	2.5	0.065	0.165
−	0.2	0.75	0.04	0.01	0.5	0.05	0.03

Notes; A: KH_2_PO_4_, B: K_2_HPO_4_, C: MgSO_4_·7H_2_O, D: CaCl·2H_2_O, E: yeast extract F: FeCl_3_·6H_2_O, G: sodium-EDTA.

**Table 7 molecules-24-02613-t007:** The CCD matrix design of five selected nutritional factors in modified BH medium.

Coded Values/Levels	Independent Variable Actual Values				
	A	B	C	D	E
−α	0.05	3.39	0.11	0.01	0.05
−1	0.60	3.80	0.80	0.02	0.60
0	1.00	4.10	1.30	0.03	1.00
+1	1.40	4.40	1.80	0.04	1.40
+α	1.95	4.81	2.49	0.05	1.95

Notes: A: PFAD, B: KH_2_PO_4_, C: yeast extract, D: sodium-EDTA, E: MgSO_4_·7H_2_O.

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
