# Peer review of "Development of Palm Fatty Acid Distillate-Containing Medium for Biosurfactant Production by Pseudomonas sp. LM19"

_molecules, 2019, doi:10.3390/molecules24142613_

Reviewer 1 Report

General comments

This manuscript presented the experimental work dealing with the optimal medium compositions for the production of biosurfactans by Pseudomonas sp. LM19 isolated from chicken processing factory. In general, I feel that this work does not contain enough scientific value nor does it provide useful information for readers in the field of biosurfactant. Hence, it is not suitable for publication in Molecules.  However, this reviewer would like to address some points as follows to assist the author in making the improvement on this manuscript.

Specific comments

(1)  What is the type of biosurfactant for the surface active compound produced from Pseudomonas sp. LM19?  The authors should be confirmed the type of biosurfactant by the analysis of chemical structure and molecular weight.

(2)  The manuscript states that the purified biosurfactant that was extracted from the culture broth of Pseudomonas sp. LM19 was a viscid product. I am curious to know how to determine the concentration of such a viscid product.

(3)  Following the preceding comment, this study determined the biosurfactant concentration by measuring the emulsion index and orcinol assay. However, these methods are not precise. The authors should review and present some precise methods, such as the HPLC method, for quantifying biosurfactant in the review section. Therefore, I strongly suggested that the author should also develop a HPLC assay method to directly measure the concentration of biosurfactant (rhamnolipid).

(4)  What is the purity of the off-white solid (rhamnolipid) collected?

(5) There are too many figures and tables. Can some of the figures or tables be combined or omitted?

Author Response

Point 1:  What is the type of biosurfactant for the surface active compound produced from Pseudomonas sp. LM19?  The authors should be confirmed the type of biosurfactant by the analysis of chemical structure and molecular weight.

Response 1: The type of biosurfactant was confirmed to be rhamnolipid as the standard used in HPLC for identification of biosurfactant class and congeners is 99% pure rhamnolipid. However, our works not analyse until chemical structure and molecular weight as our main objective is to develop or formulate good production medium to enhance biosurfactants production by Pseudomonas sp. using both combination traditional and statistical approaches.

Point 2: The manuscript states that the purified biosurfactant that was extracted from the culture broth of Pseudomonas sp. LM19 was a viscid product. I am curious to know how to determine the concentration of such a viscid product.

Response 2: Viscid product produced as in honey viscous form is not directly used for orcinol test. They have been diluted into distilled water (Line 687-688) so the sample will be analysed in liquid form and not in viscid form.

Point 3: Following the preceding comment, this study determined the biosurfactant concentration by measuring the emulsion index and orcinol assay. However, these methods are not precise. The authors should review and present some precise methods, such as the HPLC method for quantifying biosurfactant in the review section. Therefore, I strongly suggested that the author should also develop a HPLC assay method to directly measure the concentration of biosurfactant (rhamnolipid).

Response 3: Thank you for the suggestions. Since all the biosurfactant samples were quantified by orcinol test in this manuscript, the addition of quantification method by HPLC might cause slight different of value for both methods. However, we already added surface tension value in Table 2 to support our data in this manuscript.

Point 4:  What is the purity of the off-white solid (rhamnolipid) collected?

Response 4: The purity of rhamnolipid was not determined in this manuscript as the final form of the rhamnolipid was in viscous honey diluted with distilled water before further quantified by orcinol assay. The rhamnolipid was not dried until form of white solid.

Point 5: There are too many figures and tables. Can some of the figures or tables be combined or omitted?

Response 5: Some of the tables (originally Tables 6, 7 and 8) have been omitted from the manuscripts and put into supplementary data.

Reviewer 2 Report

The manuscript describes the optimization of culture medium for biosurfactant production, using a residue from palm industry, palm fatty acid distillate. The authors use a strain of Pseudomonas sp. to produce biosurfactant. I think they should justify the novelty of their work in the introduction section, since  Pseudomonas are well described for that purpose, as well as the use of residual substrates. The optimization tool used is also well described.

Regarding nitrogen sources (Type of nitrogen sources), since experiments were conducted in flasks, how your results may be influenced by oxygen availability? I think this topic lacks discussion.

The maximum biosurfactant concentration reported by the authors was around 2 g/L. There are reports in the literature with much higher titer, how do the authors explain those differences. The authors should also report those values, not only results lower than that they have achieved.

Last but not leaste, why the authors did not determine surface tension?

Minor points.

- English language should be revised, some sentences are too long and sometimes, the verb is not well used.

- In table 2, it would be interesting to report growth in the same magnitude, for instance CFUx10-8 . It is an easier way to see that medium LB favored growth.

- The topic “2.5.3. Verification of experiment” is the model validation?

- Figure 3, x-axis should begin at zero

- The topic “Identification of biosurfactants class and congeners” lacks discussion, since the authors only described the different classes they obtained. Why is this information important? How the use of the bioproduct would be influenced by its composition.

Author Response

Point 1:  The manuscript describes the optimization of culture medium for biosurfactant production, using a residue from palm industry, palm fatty acid distillate. The authors use a strain of Pseudomonas sp. to produce biosurfactant. I think they should justify the novelty of their work in the introduction section, since  Pseudomonas are well described for that purpose, as well as the use of residual substrates. The optimization tool used is also well described.

Response 1: The novelty of this project was explained in Introduction section (Line 79-83)

Point 2: Regarding nitrogen sources (Type of nitrogen sources), since experiments were conducted in flasks, how your results may be influenced by oxygen availability? I think this topic lacks discussion.

Response 2: The discussion on oxygen availability towards our results was added in Line 253-260.

Point 3: The maximum biosurfactant concentration reported by the authors was around 2 g/L. There are reports in the literature with a much higher titer, how do the authors explain those differences. The authors should also report those values, not only results lower than that they have achieved.

Response 3: The higher titer which using another substrate like sunflower oil, palm oil and olive mill waste (Line 461-467) was discussed.

Point 4:  Last but not least, why the authors did not determine surface tension?

Response 4: The surface tension data was added into the manuscript (Table 2).

Point 5: English language should be revised, some sentences are too long and sometimes, the verb is not well used

Response 5: The manuscripts had been revised using Grammarly. The changes were made in Line 14, 20, 21, 40, 44, 45, 57, 60, 78, 80, 91, 119, 121, 130, 131, 137, 138, 145, 153, 156, 158, 159, 162, 204, 205, 208, 209, 211, 213, 214, 215, 216, 217, 218, 239, 242, 246, 251, 258, 263, 270, 271, 272, 275, 287, 295, 297, 308, 321, 361, 362, 363, 380, 386, 389, 391, 392, 408, 419, 421, 422, 424, 426, 425, 427, 428, 431, 438, 449, 469, 475, 476, 480, 483, 492, 494, 499, 504, 507, 509, 511, 512, 513, 517, 519, 537, 554, 555, 557, 560, 572, 581, 600, 601, 616, 622, 663, 668, 675, 676, 687, 691, 703, 707, 713, 714, 721, 723, 724.

Point 6: In table 2, it would be interesting to report growth in the same magnitude, for instance CFUx10-8. It is an easier way to see that medium LB favoured growth.

Response 6: The magnitude for cell growth in Table 2 had been uniformed into CFUx10-8.

Point 7: The topic “2.5.3. Verification of experiment” is the model validation?

Response 7: Change the word ‘verification’ to ‘validation’ (Line 425).

Point 8:  Figure 3, x-axis should begin at zero

Response 8: Figure 3 had been corrected. X-axis all begin at zero.

Point 9: The topic “Identification of biosurfactants class and congeners” lacks discussion, since the authors only described the different classes they obtained. Why is this information important? How the use of the bioproduct would be influenced by its composition.

Response 9: The important of the information under topic “Identification of biosurfactants class and congeners” was explained (Line 500-514).

Reviewer 3 Report

The research with the title of "Development of Palm Fatty Acid Distillate-Containing Medium for Biosurfactant Production by Pseudomonas sp. LM19" is very interesting. It has a high novelty regarding the production of an important class of compunds from the industrial point of view – biosurfactants. In this study, the influence of several parameters on the biosurfactant production has been systematically evaluated. I suggest that the present manuscript is accepted for publication. However, some minor changes is required.

- Figure 1 is hard to read (low resolution). The resolution should be strongly improved.

- Tables should be presented in same page.

Author Response

Point 1:  The research with the title of "Development of Palm Fatty Acid Distillate-Containing Medium for Biosurfactant Production by Pseudomonas sp. LM19" is very interesting. It has a high novelty regarding the production of an important class of compunds from the industrial point of view – biosurfactants. In this study, the influence of several parameters on the biosurfactant production has been systematically evaluated. I suggest that the present manuscript is accepted for publication. However, some minor changes is required.

Response 1: Thank you. Some other minor changes in introduction part and discussion part were done (Line 80-83, 253-260, 461-467, 471-474, 500-514).

Point 2: Figure 1 is hard to read (low resolution). The resolution should be strongly improved.

Response 2: The resolution for Figure 1 had been improved to higher 600 dpi.

Point 3: Tables should be presented in same page.

Response 3: Table 4 was compressed to make it fit into one page.

Reviewer 4 Report

The manuscript describes the development of palm fatty acid  as carbon source for rhamnolipid production. However, such studies are not new and the results obtained are quite normal as a case study. Also, the response surface methodology method used in the present study is an too old and traditional method to add significant value to the existing knowledge in this area. Hence, the manuscript is not acceptable in the present style as a full paper.

1. Line 22: 1.148%  (v/v) PFAD; Line 671 : 1.148 g/L PFAD;
2. the production yield(2.424g/L) and productivity(0.3463g/L/d) were relatively low compared those with plant oil (palm oil)as  carbon source, the authors should focus on and discuss this result.
3.  the production yield(2.424g/L) and Y p/s (0.27g/g)were higher than those in reference 13, due to  the synthetic ability of strains and the compositional profile of PFAD ?
4. the compositional profile of PFAD were long chain fatty acids ranging from C12-C18, whether the cells have a preference for substrates?

Author Response

Point 1:  The manuscript describes the development of palm fatty acid as carbon source for rhamnolipid production. However, such studies are not new and the results obtained are quite normal as a case study. Also, the response surface methodology method used in the present study is an too old and traditional method to add significant value to the existing knowledge in this area. Hence, the manuscript is not acceptable in the present style as a full paper.

Response 1: This study comprised of the usage of rare substrate but abundant produced after the refinery process in palm oil industry. To our knowledge, this is second work utilized PFAD as substrate for biosurfactant production (first is by Nazren Radzuan et al., 2017) for purpose to add value to it. In the previous study, it involved the comparison of biosurfactant production by Pseudomonas sp. using glucose and PFAD as substrate. Since the biosurfactant production is almost similar for both substrates, this study attempted to improve the production in term of nutritional factors (both macro and micronutrients). We combined both old (one-factor-at-time) and statistical techniques (PBD and RSM) in developing good production medium. We are not depending only on RSM to formulate this medium. Furthermore, the second part of this work involved in biosurfactant class and their congeners. The importance of this part of work was explained in Line 500-514. Since Table 5 showing there’s difference between the congeners produced when using different substrate (PFAD and PKFAD) which having different free fatty acid content, further work can be done by experimenting different type of free fatty acid as substrate which might influence the congeners produced later. Last but not least, this paper cover wide range of work starting from the substrate selection until identifying the end product produced.

Point 2: Line 22: 1.148% (v/v) PFAD; Line 671: 1.148 g/L PFAD;

Response 2: The correct one was 1.148% (v/v) and Line 671 was corrected.

Point 3: The production yield (2.424g/L) and productivity (0.3463g/L/d) were relatively low compared those with plant oil (palm oil) as carbon source, the authors should focus on and discuss this result.

Response 3: The discussion of palm oil as substrate as compared to our study was added in Line 461-467.

Point 4: The production yield (2.424g/L) and Yp/s (0.27g/g) were higher than those in reference 13, due to the synthetic ability of strains and the compositional profile of PFAD? 

Response 4: The difference of yield might first be due to the different composition of free fatty acid (FFA) supplied into the culture broth. Secondly, even both are coming from Pseudomonas sp., the strain was different which lead to the assumption in which biosurfactant production is also strain dependent. It was explained in Line 471-474.

Point 5: The compositional profile of PFAD were long chain fatty acids ranging from C12-C18, whether the cells have a preference for substrates?

Response 5: We are not performing those experiments in this part of study. However, there are some articles showing there’s potential for different microbes of having preference for different free fatty acids in producing their metabolism as in reference 43. In future, we are really interested to explore about it. 

Round  2

Reviewer 1 Report

The manuscript has been revised closely according to my comments.  Hence, it should be suitable for publication in Molecules.

Reviewer 4 Report

The authors improved the manuscript according to the reviewer' comments. I think the quality of the manuscript meets the requirement of publication.